**Investigation**

# Complex epistatic interactions between ELF3, PRR9, and PRR7 regulate the circadian clock and plant physiology

Li Yuan [iD],[1,†] Paula Avello,[2,3,†] Zihao Zhu [iD],[4,†] Sarah C.L. Lock [iD],[5,†] Kayla McCarthy,[5] Ethan J. Redmond,[5] Amanda M. Davis,[5] Yang Song [iD],[1] Daphne Ezer [iD],[5] Jonathan W. Pitchford,[2,5] Marcel Quint [iD],[4] Qiguang Xie [iD],[1] Xiaodong Xu [iD],[1] Seth J. Davis [iD],[1,5,*] James Ronald [iD][5,6,*†]

[1]State Key Laboratory of Crop Stress Adaptation and Improvement, School of Life Sciences, Henan University, Kaifeng 475004, China
[2]Department of Mathematics, University of York, York, YO10 5DD, UK
[3]School of Biology, Faculty of Biological Sciences, University of Leeds, Leeds LS2 9JT, UK
[4]Institute of Agricultural and Nutritional Sciences, Martin Luther University Halle-Wittenberg, Halle (Saale) 06108, Germany
[5]Department of Biology, University of York, Wentworth Way, York, YO10 5DD, UK
[6]School of Molecular Biosciences, College of Medical, Veterinary and Life Sciences, University of Glasgow, Bower Building, University Avenue, Glasgow G12 8QQ, UK

*Corresponding author: Email: seth.davis@york.ac.uk; *Corresponding author: Email: james.ronald@glasgow.ac.uk
[†]These authors contributed equally.

Circadian clocks are endogenous timekeeping mechanisms that coordinate internal physiological responses with the external environment. EARLY FLOWERING3 (ELF3), PSEUDO RESPONSE REGULATOR (PRR9), and PRR7 are essential components of the plant circadian clock and facilitate entrainment of the clock to internal and external stimuli. Previous studies have highlighted a critical role for ELF3 in repressing the expression of *PRR9* and *PRR7*. However, the functional significance of activity in regulating circadian clock dynamics and plant development is unknown. To explore this regulatory dynamic further, we first employed mathematical modeling to simulate the effect of the *prr9/prr7* mutation on the *elf3* circadian phenotype. These simulations suggested that simultaneous mutations in *prr9/prr7* could rescue the *elf3* circadian arrhythmia. Following these simulations, we generated all *Arabidopsis elf3/prr9/prr7* mutant combinations and investigated their circadian and developmental phenotypes. Although these assays could not replicate the results from the mathematical modeling, our results have revealed a complex epistatic relationship between ELF3 and PRR9/7 in regulating different aspects of plant development. ELF3 was essential for hypocotyl development under ambient and warm temperatures, while PRR9 was critical for root thermomorphogenesis. Finally, mutations in *prr9* and *prr7* rescued the photoperiod-insensitive flowering phenotype of the *elf3* mutant. Together, our results highlight the importance of investigating the genetic relationship among plant circadian genes.

**Keywords:** circadian clock; plant development; thermomorphogenesis

## Introduction

The daily rotation of Earth generates predictable *diel* cycles in light and temperature. Circadian clocks are molecular timekeeping mechanisms that anticipate these daily oscillations, allowing physiological responses to be coordinated with the external environment. This anticipatory ability is dependent on the circadian system undergoing daily re-setting in response to stimuli with predictable oscillatory patterns, a process called entrainment. In plants, light at dawn is thought to be a major entrainment signal (Millar 2004). However, temperature, humidity, and sugar availability also function in the entrainment of the plant circadian clock (Webb *et al.* 2019). The circadian clock has a central role in the life history of plants, regulating germination, vegetative and floral development, metabolism, and the response to biotic and abiotic stress. As such, plants whose circadian clock is not closely aligned with the external environment have reduced fitness (Dodd *et al.* 2005; Greenham and McClung 2015; Xu *et al.* 2022).

The plant circadian clock is an interconnected regulatory network of transcriptional, translational, and posttranslational feedback loops (McClung 2019). Genetic and biochemical studies over recent decades have identified more than 20 different components that are involved in the circadian oscillator. Recent mathematical modeling has worked to reduce this complexity, resulting in a compact model that describes 8 genes condensed into 4 components: CIRCADIAN CLOCK ASSOCIATED1 (CCA1) and LATE ELONGATED HYPOCOTYL (LHY) termed CL, PSEUDO RESPONSE REGULATOR9 (PRR9) and PRR7 termed P97, TIMING OF CAB EXPRESSION1 (TOC1) and PRR5 termed P51, and EARLY FLOWERING4 (ELF4) and LUX ARRYTHMO (LUX) termed EL (De Caluwé *et al.* 2016). Modifications of this simplified model have been used to understand the effect of external and internal cues on the circadian system and spatial differences in the plant circadian clock (Ohara *et al.* 2018; Avello *et al.* 2019; Greenwood *et al.* 2022).

In these compact models, the EL component describes the evening complex (EC). The EC is a tripartite protein complex composed of EARLY FLOWERING3 (ELF3), ELF4, and LUX (Nusinow *et al.* 2011; Herrero *et al.* 2012). LUX is a transcription factor that

is necessary for the recruitment of ELF3 and ELF4 onto chromatin (Nusinow *et al.* 2011). ELF3 recruits chromatin remodeling enzymes including histone deacetylases, histone demethylases, and nucleosome remodeling complexes to repress gene expression (Lee *et al.* 2019; Park *et al.* 2019; Tong *et al.* 2020; Lee and Seo 2021). The role of ELF4 within the EC remains unresolved but may facilitate the nuclear localization of ELF3 and separately facilitate the binding of the EC to DNA (Kolmos *et al.* 2011; Herrero *et al.* 2012; Anwer *et al.* 2014; Silva *et al.* 2020; Ronald *et al.* 2021). Mutations in a single *ec* components result in circadian arrhythmia, although the role and importance of the EC in regulating circadian rhythms in root cells are still uncertain (Covington *et al.* 2001; Doyle *et al.* 2002; Onai and Ishiura 2005; Chen *et al.* 2020; Nimmo *et al.* 2020). In addition to regulatory activity in the oscillator, ELF3 is also necessary for light and temperature entrainment (Anwer *et al.* 2020; Zhu *et al.* 2022). ELF3 functions independently of the other EC components in mediating temperature entrainment (Zhu *et al.* 2022), while the requirement of the EC in facilitating ELF3-mediated light entrainment remains to be investigated.

A key target of the EC within the circadian clock is *PRR9* and *PRR7* (Kolmos *et al.* 2011; Herrero *et al.* 2012). PRR9 and PRR7 are transcription factors that share partial functional overlap within the circadian clock, with the *prr9/prr7* mutant having a longer circadian period than the respective single mutants (Farré *et al.* 2005; Salomé and McClung 2005). *PRR7* and *PRR9* along with *PRR5* also share genetic redundancy in flowering time and hypocotyl elongation (Nakamichi *et al.* 2005). Both PRR9 and PRR7 function in the entrainment pathways of the circadian clock. The expression of *PRR9* is responsive to red light and this responsiveness is regulated by ELF3 (Farré *et al.* 2005; Ronald *et al.* 2022). *PRR7* expression is regulated by sugar availability and this regulation underpins sucrose entrainment of the oscillator (Frank *et al.* 2018). *PRR9* and *PRR7* also facilitate temperature entrainment, though the nature of this entrainment pathway remains to be investigated (Salomé and McClung 2005). The EC directly regulates *PRR9* and *PRR7* expression, and accordingly, *elf3* mutants have elevated expression of *PRR9* and *PRR7* (Thines and Harmon 2010; Kolmos *et al.* 2011; Herrero *et al.* 2012). At dusk, LUX binds to the promoter of both genes and ELF3 recruits the SWI2/SNF2-RELATED (SWRI) histone-remodeling complex to induce a repressive chromatin state at both loci (Tong *et al.* 2020). Previous mathematical and genetic analysis indicates a stronger repressive effect of ELF3 on *PRR9* than *PRR7* (Herrero *et al.* 2012), although the significance of this remains unclear.

Although the expression of *PRR9* and *PRR7* is constitutively increased to a highly elevated level in the *elf3* mutant background (Kolmos *et al.* 2011), the importance of this misexpression in contributing to the *elf3* circadian and physiological phenotypes has yet to be investigated. Here, through the use of mathematical modeling, molecular assays, and physiological measurements, we sought to test the importance of the individual and high-order mutations in *prr9* and *prr7* in contributing to the different *elf3* phenotypes. Together, our results have highlighted a complex epistatic interaction between ELF3, PRR9, and PRR7.

## Materials and methods
### Plant lines
All mutant lines described in this work are in the Col-0 background. The Col-0, *prr9-1*, *prr7-3*, and *prr9-1/prr7-3* mutant lines harboring *CCA1::LUC* have all been described previously (Farré *et al.* 2005). The new mutant combinations generated in this work were created by crossing the *elf3-1* (Zagotta *et al.* 1996)

*CCA1::LUC* mutant into the *prr9-1/prr7-3* double mutant. The various single, double, and triple mutant combinations were then identified through genotyping.

## Modeling
The original De Caluwé model considered *PRR9* and *PRR7* having similar expression profiles and they were characterized as a single model variable (P97) (De Caluwé *et al.* 2016). Here, we modified the DC Caluwé model to separate the variable P97 so that 3 models were implemented:

- Model 1: The role of each separated component within the network, as described by De Caluwé *et al.* 2016, was unchanged.
- Model 2: The interaction between CCA1/LHY (CL) and PRR9 (P9) was modified; PRR9 is inhibited by CCA1/LHY.
- Model 3: The interaction between CCA1/LHY (CL) and PRR9 and PRR7 was modified; PRR9 and PRR7 are now inhibited by CCA1/LHY. Also, a negative autoregulation in CCA1/LHY was introduced as described in Greenwood *et al.* (2022).

The resulting models consist of 11 ordinary differential equations (ODEs) to reproduce responses to light, rather than 9 as in De Caluwé *et al.* 2016. Parameter values were taken from the original model. In model 2, where *PRR9* is repressed by *CCA1/LHY*, the mean of the original parameter values that go into P97 inhibition was used for parameterization; that is, the new parameter incorporated into the model is the mean between the Hill function parameters $K_4$ and $K_5$ (De Caluwé *et al.* 2016) for inhibition of *PRR9/PRR7* by *PRR5/TOC1* and *ELF4/LUX* in the original model, respectively. We name this parameter as $K_{11}$. The added equations are

$$\frac{d[P9]_m}{dt} = (v_{2L}*L*[P]+v_{2A})*\frac{1}{1+\left(\frac{[P51]_p}{K_4}\right)^2+\left(\frac{[EL]_p}{K_5}\right)^2+\left(\frac{[CL]_p}{K_{11}}\right)^2}-k_2*[P9]_m,$$

$$\frac{d[P9]_p}{dt} = p_2*[P9]_m-(d_{2D}*D+d_{2L}*L)*[P9]_p.$$

Additionally, we implemented the U2019.3 model for comparison (Urquiza-García and Millar 2021). This model is governed by the ODEs presented in Pokhilko *et al.* (2012) but was parameterized using a different approach with the aim to obtain simulated data in absolute units (Urquiza-García and Millar 2021).

Simulations were carried out using experimental conditions, i.e. 8 days of entrainment at 12 hours light and 12h dark in a 24-h cycle. The clock then was released into constant light conditions. Null mutants were simulated by setting the relevant rate constant of transcription to 0 ($v_{2A}$ and $v_4$ in De Caluwé *et al.* 2016; n4, n7, n8, n9, and n3 in Urquiza and Millar 2021). All simulations were done in MATLAB.

## Luciferase circadian experiments
Seeds were surface sterilized before being sown on 1× MS plates with 3% *w/v* sucrose, 0.5 g/L MES, and a pH of 5.7. Seeds were then stratified for 3 days. After 3 days, plates were moved to a neutral-day (ND) photoperiod chamber with a set temperature of 22°C and a light intensity of 85 μmol/m$^{-2}$/s$^{-1}$ for 5 days. On day 6, seedlings were transferred to a black 96-microwell plate containing the same media as described before. 15-μL 5 mM

luciferin was then added superficially to the media. Seedlings were then returned to the same entrainment chamber for 24 h. After 24 h of re-entrainment, seedlings were transferred to the TopCount before subjective dusk. All TopCount experiments were carried out under constant red (3 μmol/m$^{-2}$/s$^{-1}$) and blue light (1.1 μmol/m$^{-2}$/s$^{-1}$). Data analysis was performed as described previously (Kolmos et al. 2009).

## Gene expression analysis

Comparative analysis of selected genes was performed by quantitative real-time PCR (qRT-PCR). Seedings were grown in a plant growth chamber (Percival Scientific, Model CU-36L5) at 22°C under 12:12 long-day (LD) cycles (white light, 100 μmol/m$^{-2}$/s$^{-1}$) for 8 days and then transferred into constant light and temperature conditions. Samples were harvested and frozen at 3h intervals. Total RNA was isolated using RNAiso Plus (TaKaRa, Cat. #9109) and reverse transcribed to produce cDNA using the RevertAid First Strand cDNA Synthesis Kit (Thermo Fisher Scientific, Cat. #K1622) following the manufacturer's protocol. The qRT-PCR was performed by using the TB Green Premix Ex Taq (Tli RNase H Plus; TaKaRa, Cat. #RR420A) and CFX Connect Real-Time System (Bio-Rad). Primer sequences used for qRT-PCR are listed in Supplementary Table 1.

## Flowering time measurements

Flowering time experiments were carried out as described previously (Kolmos et al. 2011). In brief, seeds were surface sterilized and sown onto 1× MS plates with 0.25% w/v sucrose, 0.5 g/L MES, and a pH of 5.7 and stratified for 3 days at 4°C. Plates were then moved to a ND photoperiod chamber with a set temperature of 22°C and a light intensity of 85 μmol/m$^{-2}$/s$^{-1}$ for 14 days. After 14 days, seedlings were transferred to soil and moved to a short-day (SD) or LD photoperiod. Under both photoperiods, the temperature was set to 22°C and light intensity of 85 μmol/m$^{-2}$/s$^{-1}$. Flowering was determined as the point at which the inflorescence was ~1 cm above the rosette. For each genotype, 15–20 plants were analyzed. All experiments were repeated twice with similar results observed each time.

## Hypocotyl and root measurements

Seeds were surface sterilized and plated on solid A. thaliana solution medium (Lincoln et al. 1990) with 1% w/v sucrose. Seeds were then stratified for 3 days before being transferred to a SD chamber (8 h of light and 16 h of darkness) with a light intensity of 90 μmol/m$^{-2}$/s$^{-1}$. The temperature the seedlings were exposed to is described in the text. Seedlings were allowed to grow for 8 days before scans of the plates were taken. The same scan was used to measure hypocotyl and root growth. Measurements were calculated using ImageJ. A minimum of 42 seedlings were measured per genotype across 2 experimental repeats.

## Statistical analysis

To determine the statistical difference for the physiological experiments (Figs. 3–5), an ANOVA with a Tukey honestly significance difference (HSD) post hoc test was performed. The alpha level (P-value) for the post hoc test was set to 0.05 to determine the significant difference between the genotypes and conditions. All statistical analyses were performed in Rstudio (v.2022.07.02) with the version 3.6.1 of R.

# Results

## Modeling suggests a role of PRR9/PRR7 in the *elf3* arrhythmia phenotype

To provide insights into the possible role of PRR9/PRR7 in contributing to the circadian phenotype of *elf3*, we first simulated the effects of the *prr9* and *prr7* mutations on the *elf3* circadian phenotype using the compact DC2016 model (De Caluwé et al. 2016). ELF3 is not implicitly modeled in the DC2016 model. Instead, the activity of ELF3 is represented by LUX and ELF4 within the EL component. *elf4* and *lux* mutants are similarly arrhythmic to *elf3* (Doyle et al. 2002; Onai and Ishiura 2005). Hence, we will use simulated mutations in *el* as a proxy for mutations in *elf3*.

In the original compact model (DC2016), the *PRR9* (*P9*) and *PRR7* (*P7*) genes are grouped together in 1 component termed P97 (De Caluwé et al. 2016). Thus, it was firstly necessary to separate the P97 component into 2 components termed *P9* and *P7* (see *Materials and Methods* for further details). Three different models were then implemented; in the first model, the individual P9 and P7 components retained the original functions of the P97 component as described in the DC2016 model. In the second model, we introduced a negative regulatory connection from CL (CCA1/LHY) to P9. In the third model, a negative regulatory connection from CL to P9 and P7 was introduced along with a negative autoregulation in CL as described previously (Greenwood et al. 2022) (Fig. 1a). The modifications were made to reflect CL now being described as repressors of *PRR* expression (Adams et al. 2015).

To understand whether the 3 different models could accurately simulate the *elf3/prr* phenotype, we firstly simulated the expression of *CL* in wild type (WT) and mutants with well-defined circadian phenotypes. All 3 models were firstly conditioned for 8 days of entrainment under ND photoperiods before being released into constant white light. The simulated expression of *CL* peaked at dawn for WT in model 1 and model 2, which is consistent with experimental data (Fig. 1b and d) (Kolmos et al. 2011). The output of *CL* in the simulated *elf3*, *prr9*, *prr7*, and *prr9/prr7* mutants also closely replicated the reported expression profile of *CCA1/LHY* in the respective backgrounds for both model 1 and model 2 (Farré et al. 2005; Kolmos et al. 2011) (Fig. 1b and d; Supplementary Fig. 1a). In contrast, the expression of *CL* in model 3 did not replicate the reported expression profile of *CCA1/LHY* in WT or the different mutants (Supplementary Fig. 2a). A similar behavior was also observed for *P51* (*PRR5/TOC1*). Again, the outputs of model 1 and model 2 closely followed the reported expression profile of *P51* for WT and the simulated mutants (Fig. 1c and e; Supplementary Fig. 1b) (Kolmos et al. 2011), while model 3 did not accurately reflect the expression of *P51* in any instance (Supplementary Fig. 2b). Therefore, we will not further discuss the outputs of model 3 in this work.

We simulated the effect of combining the *elf3* and *prr9*, *prr7*, *prr9/prr7* mutations on the expression of *CL* and *P51* under free-running conditions. As with the *elf3* single mutant, the expression of *CL* was rhythmic in the *elf3/prr9* and *elf3/prr7* mutants under photocycles but became arrhythmic upon release into free-running conditions (Fig. 1b and d). A similar behavior was observed for the output of *P51*, with the expression of *P51* quickly becoming arrhythmic upon release into free-running conditions for *elf3/prr9* and *elf3/prr7* (Fig. 1c and e). In contrast, the expression of *CL* and *P51* remained rhythmic under free-running conditions in the simulated *elf3/prr9/prr7* mutant. Interestingly, the expression of *P51* and *CL* peaked earlier in the *elf3/prr9/prr7* than in WT in model 1 and model 2 (Fig. 1b–d). Previous work has suggested that an extremely early phase in the *elf3* mutant may explain the *elf3* arrhythmic phenotype (Kim et al. 2005). Together, the

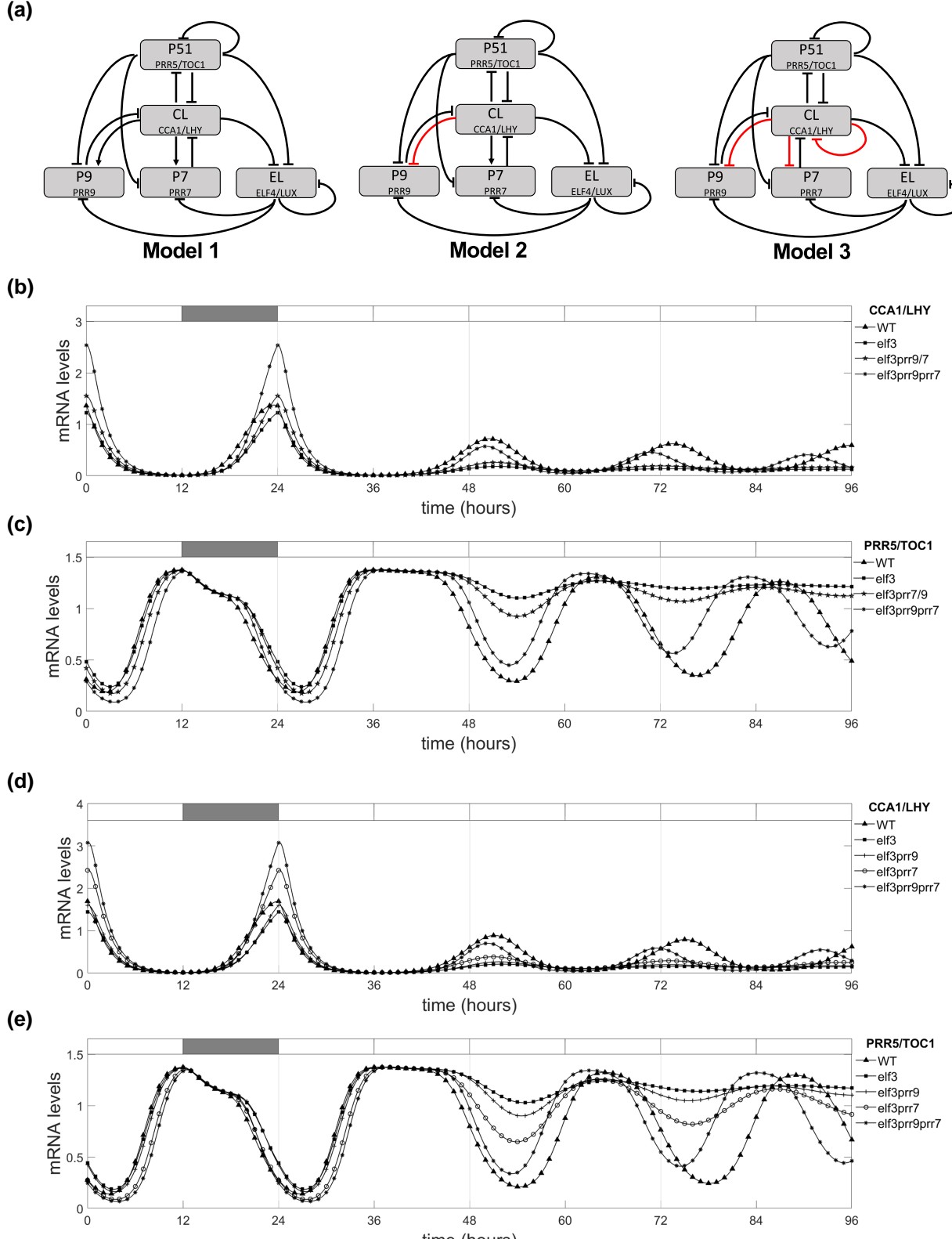

**Fig. 1.** Modeling suggests that the *prr9/prr7* may contribute to the *evening complex* circadian arrhythmia. a) Three modifications of the De Caluwé *et al.* 2016 (DC2016) model presenting *PRR9* and *PRR7* as separate components. Alterations to the original DC2016 model are highlighted in red. Model 1 is the original layout described in DC2016 but with P9 and P7 split into 2 components. In model 2, a negative interaction is introduced from CL to P9, while in model 3, a negative interaction is introduced from CL to P9 and P7, and negative autoregulation of CL. Outputs from model 1 for the expression of b) *CL* and c) *P51* and outputs from model 2 for the expression of d) *CL* and e) *P51* in the WT and simulated *elf3/prr* mutant backgrounds are shown. For model 1, the outputs for the *elf3/prr9* and *elf3/prr7* are represented as a single output (*elf3prr9/7*) as model 1 keeps the same functions for the respective mutants.

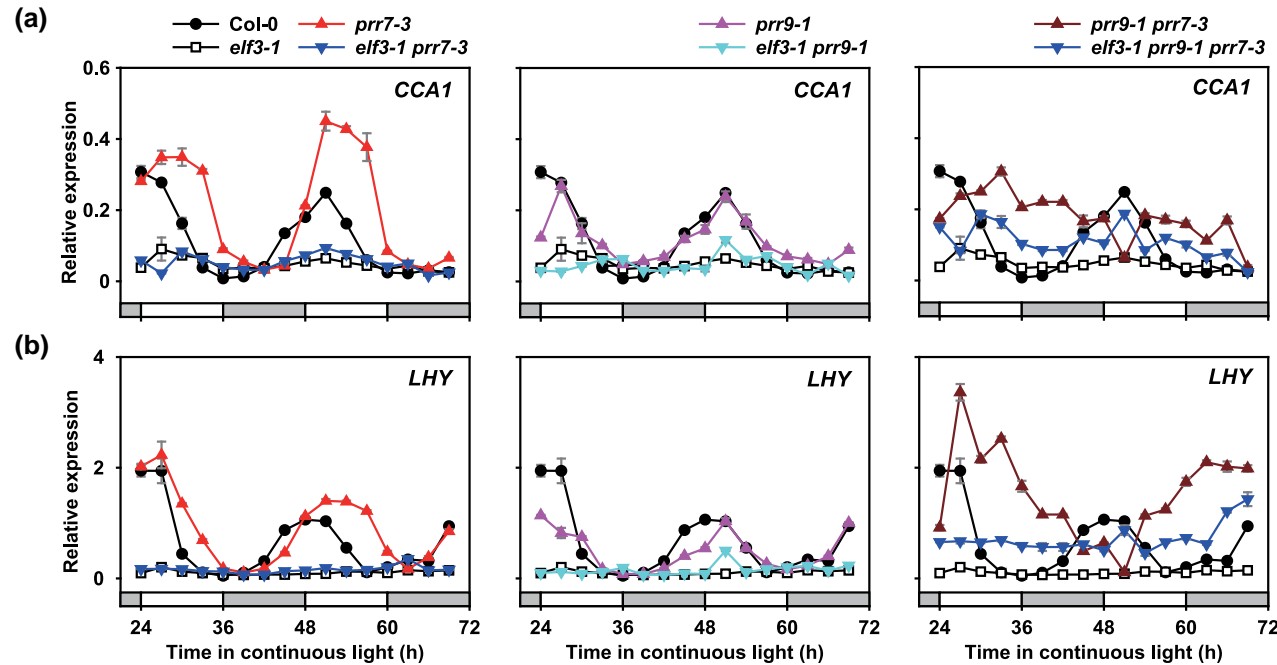

**Fig. 2.** *Prr9/prr7* does not rescue the arrhythmicity of *CCA1* or *LHY* expression in the *elf3* background. The expression of a) *CCA1* or b) *LHY* in WT (Col-0), the *elf3*, *prr*, and *elf3/prr* mutant backgrounds under constant light and constant temperature of 22°C. Seedlings were prior entrained under 12:12 light–dark cycles with a set temperature of 22°C for 8 days. Data are the mean of 3 technical replicas; error bars represent standard error. All experiments were repeated twice, with similar results observed each time. *IPP2* was used as a normalization control. White or gray bars represent subjective day and subjective night.

output of these models suggests a role of PRR9 and PRR7 in the arrhythmicity of the *elf3* mutant.

## The *prr9/prr7* mutations do not restore rhythmicity in the *elf3* mutant background

As the outputs of the models indicate that PRR9/PRR7 may contribute to the arrhythmicity of the *elf3* mutant, we generated the different *elf3/prr9/prr7* mutant combinations to investigate this possibility. The *prr9-1/prr7-3* (*prr9/prr7* henceforth) double mutant was crossed into the *elf3-1 CCA1::LUC* (*elf3* henceforth) background to generate all *elf3/prr9/prr7* single, double, and triple combinations with the *CCA1::LUC* reporter gene. This resulted in 8 genotypic comparisons (Fig. 2).

We first investigated the expression of *CCA1* and *LHY* in the respective mutants entrained under a ND photoperiod and then released into constant light and constant temperature. As described previously, the expression of *CCA1* peaked at dawn in WT before rapidly declining (Fig. 2a). In the *elf3* mutant, *CCA1* expression was arrhythmic and stayed suppressed below WT levels across the time course. There was no overt effect of the *prr9* mutation on *CCA1* expression, while in the *prr7* background, the expression of *CCA1* was elevated and the peak accumulation was shifted from dawn to the early morning (Fig. 2a). The peak expression of *CCA1* was further delayed in the *prr9/prr7* double mutant until the subjective afternoon. There was no noticeable effect of the *prr9* or *prr7* mutation on the *elf3* phenotype, with the respective double mutants closely resembling the *elf3* single and remaining arrhythmic (Fig. 2a). The *elf3/prr9/prr7* triple mutant was also arrhythmic, but the expression of *CCA1* in the triple mutant was partially elevated relative to *elf3*. The expression pattern of *LHY* was comparable to that observed for *CCA1* for each respective mutant (Fig. 2b). Neither the *prr9*, *prr7*, or *prr9/prr7* mutations could rescue the arrhythmicity of *LHY* in the *elf3* background

but, as before, there was a partial increase in the expression of *LHY* in the *elf3/prr9/prr7* triple mutant (Fig. 2b). To confirm that neither the *prr9*, *prr7*, nor *prr9/prr7* mutant could rescue the *elf3* arrhythmic phenotype, we analyzed the output of the *CCA1::LUC* reporter. As with gene expression, all *elf3/prr* combinations were arrhythmic under free-running conditions (Supplementary Fig. 4). In summary, our data suggest that the misexpression of *PRR9* and *PRR7* cannot by themselves explain the arrhythmicity of the *elf3* mutant.

## ELF3 functions downstream of PRR9/7 in controlling ambient and warm temperature-induced hypocotyl elongation

Circadian regulation of hypocotyl elongation primarily occurs through the regulation of *PHYTOCHROME-INTERACTING FACTOR4* (*PIF4*)/*PIF5* expression, stability, and transcriptional activity (Favero *et al.* 2021). Recently, ELF3 and PRR5/TOC1 were described to coordinately regulate the expression of *PIF4/5* in controlling hypocotyl elongation under SD and LD photoperiods (Li *et al.* 2020). Separate studies have also highlighted a role for PRR9/7 in regulating hypocotyl elongation via PIFs (Martín *et al.* 2018). Therefore, we tested whether there was a similar additive effect of the *elf3/prr9/prr7* mutation on hypocotyl elongation as described for the *elf3/toc1/prr5* mutant.

Firstly, we analyzed the hypocotyl phenotype of the *elf3/prr* mutants under SD photoperiods with an ambient temperature of 20°C (Fig. 3; Supplementary Fig. 5). As reported previously, the *elf3* mutant had a long hypocotyl phenotype under these conditions compared to WT Col-0 (Fig. 3a). There was no discernible hypocotyl phenotype in the *prr9* single mutant, while the *prr7* single mutant had a slightly longer hypocotyl compared to WT (Fig. 3a). The hypocotyl of the *prr9/prr7* double mutant was further elongated than the *prr7* single mutant but remained shorter than

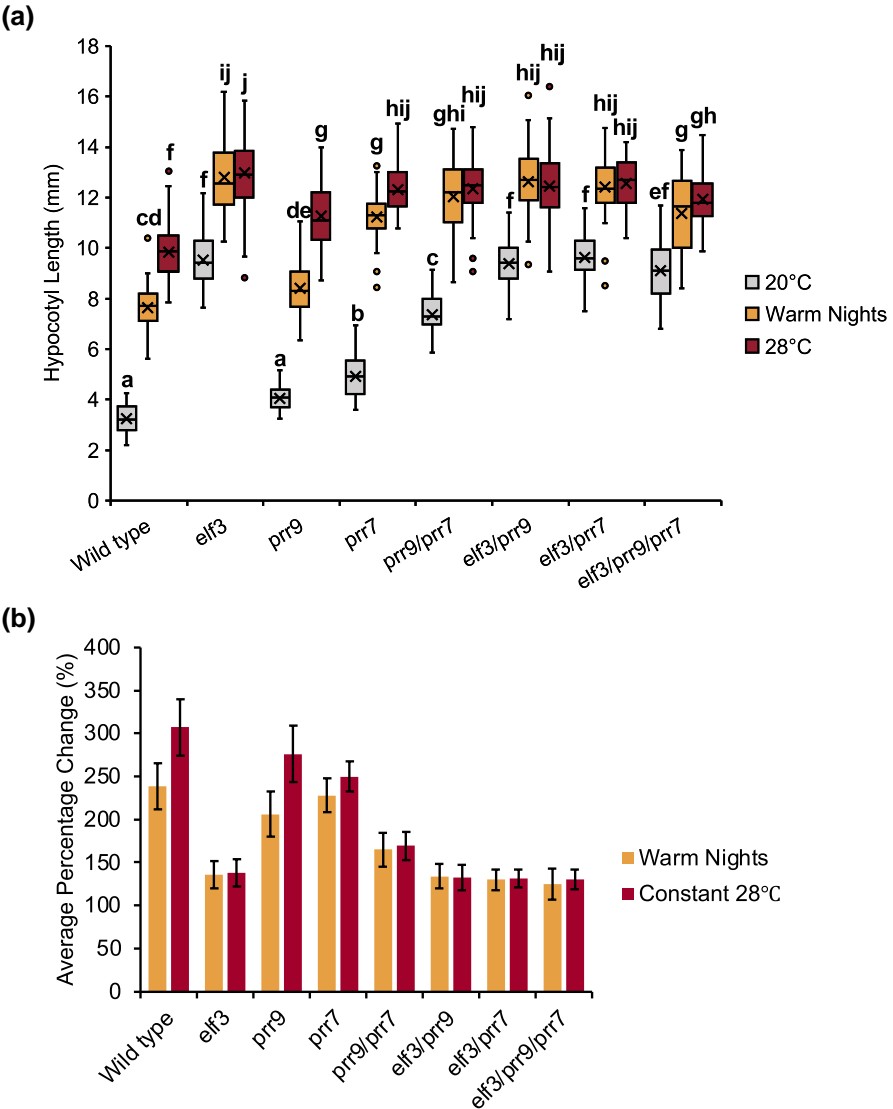

**Fig. 3.** ELF3 functions downstream of PRR9/PRR7 in controlling hypocotyl development under ambient and elevated temperatures. a) Hypocotyl length of WT (Col-0) and the different *elf3*, *prr9*, and *prr7* mutants under SD photoperiods. Seedlings were grown with an ambient temperature of 20°C (gray), 28°C warm nights (orange), or constant 28°C (red) temperatures. b) Average percentage change under warm nights or constant 28°C for the different genotypes. Error bars represent standard deviation. Experiments were repeated twice, with the presented data a combination of the 2 experiments. A minimum of 40 seedlings were analyzed. Letters signify a significant difference of $P < 0.05$, determined by ANOVA with a Tukey HSD post hoc test.

the *elf3* single mutant. Notably, there was no effect of either *prr* single or double mutations on the *elf3* phenotype, with all mutant combinations closely resembling *elf3* (Fig. 3a). Together, this suggests that PRR9 and PRR7 redundantly regulate hypocotyl elongation and function in the same pathway as ELF3 rather than working cooperatively as previously described with ELF3 and TOC1/PRR5.

Next, we investigated the *elf3/prr* mutant hypocotyl phenotypes in SD photoperiods with either warm nights (28°C exclusively during the dark phase) or constitutive exposure to 28°C. In the WT Col-0 background, exposure to a 28°C warm night was sufficient to promote hypocotyl elongation (Fig. 3a). However, growth under constant temperatures of 28°C had a stronger effect on hypocotyl elongation compared to warm nights only (Fig. 3b). The *prr9* and *prr7* single mutants had a similar response to WT, although there was a smaller difference between the constant 28°C and warm nights in the *prr7* background. Temperature responsiveness was strongly reduced in the *prr9/prr7* double mutant

and there was no longer an additive effect of constant 28°C compared to warm nights (Fig. 3b). Hypocotyl elongation in the *elf3* mutant was only weakly responsive to the elevated temperature, consistent with earlier reports (Jung *et al.* 2016; Ding *et al.* 2018). There was also no difference in response between warm nights or constitutive exposure to 28°C in the *elf3* background (Fig. 3b). A similar response was also observed in the *elf3/prr* double and triple mutants, with all combinations closely resembling the response of the *elf3* single mutant (Fig. 3a and b). Therefore, as with ambient temperature, ELF3, PRR9, and PRR7 likely function in the same pathway to control warm temperature-induced hypocotyl elongation and ELF3 functions downstream of PRR9/PRR7 in this pathway.

## PRR9 is essential for thermomorphogenic root development

Unlike in the hypocotyl, the nature and role of the circadian clock in controlling root development continue to be unclear. Recent

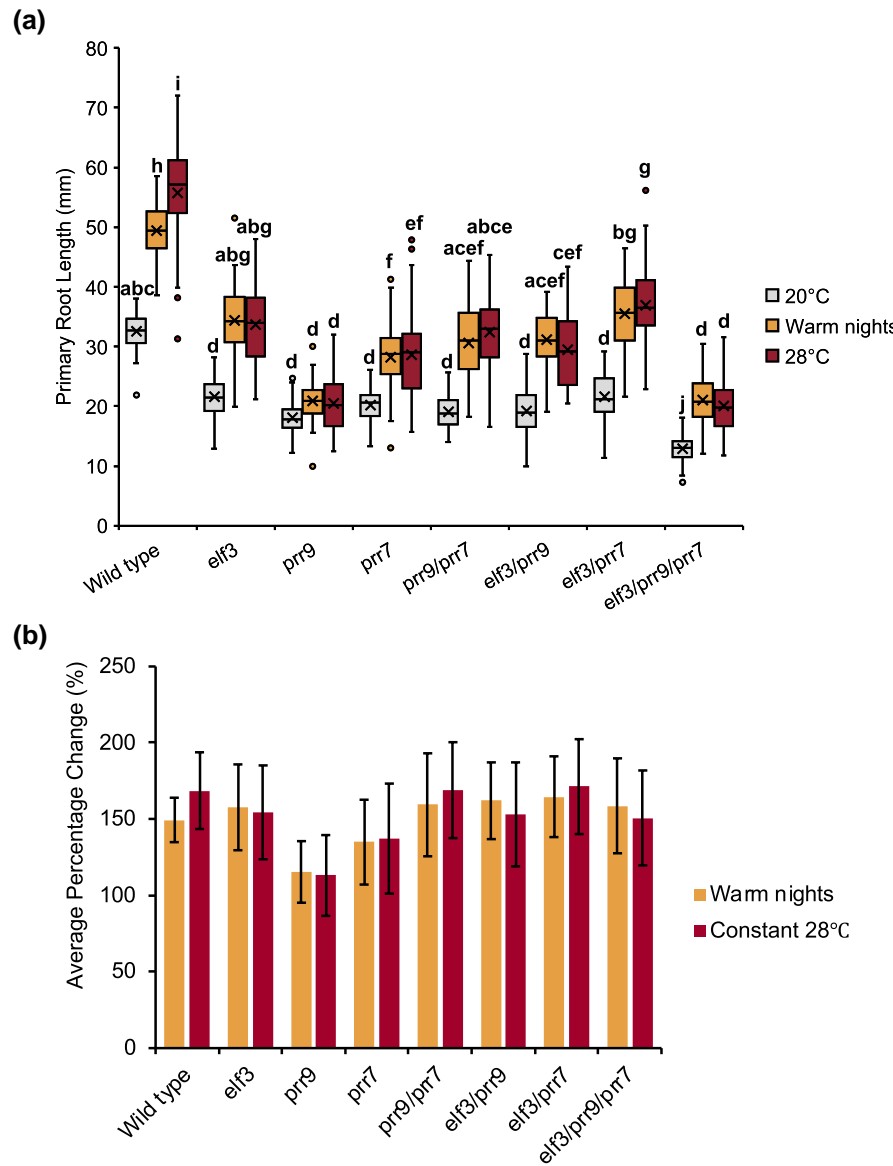

**Fig. 4.** *PRR9* is necessary for root thermomorphogenesis. a) Root length of WT (Col-0) and the different *elf3*, *prr9*, and *prr7* mutants under SD photoperiods. Seedlings were grown with an ambient temperature of 20°C (gray), 28°C warm nights (orange), or constant 28°C (red) temperatures. b) Average percentage change under warm nights or constant 28°C for the different genotypes. Error bars represent standard deviation. Experiments were repeated twice, with the presented data a combination of the 2 experiments. A minimum of 40 seedlings were analyzed. Letters signify a significant difference of $P < 0.05$, determined by ANOVA with a Tukey HSD post hoc test.

work highlighted a role for the EC in controlling lateral root development (Chen *et al.* 2020), while PRRs also regulate root development (Ruts *et al.* 2012; Li *et al.* 2019) and repress the expression of the *EC* components in root tissue under warm temperatures (Yuan *et al.* 2021). Therefore, we characterized the *elf3/prr* primary root phenotypes. As with hypocotyl development, we analyzed root growth under a SD photoperiod with either constant 20°C, 28°C warm nights only, or constant 28°C. Root development was strongly impaired in the *elf3*, *prr9*, and *prr7* single mutants at 20°C compared to WT, with each respective single mutant having a similar response to each other (Fig. 4a). There was no change in the primary root length of *prr9/prr7* double and *elf3/prr* double combinations compared to the respective single mutants. However, root development was further impaired in the *elf3/prr9/prr7* triple mutant compared to the single and double mutants (Fig. 4a). Therefore, at ambient temperatures, ELF3 and

PRR9/7 may regulate root development through separate but also partially overlapping pathways.

Exposure to warm temperature promoted root growth in WT, consistent with other reports in the literature (Quint *et al.* 2005; Hanzawa *et al.* 2013). As with hypocotyl development, constitutive exposure to 28°C caused a greater response than just warm nights, although the magnitude difference between the 2 growth conditions was smaller than observed for hypocotyl elongation (Figs. 3b and 4b). Root growth in the *elf3* single mutant remained responsive to warmth (Fig. 4a) but there was no difference in the magnitude of response between constant exposure to 28°C or warm nights only (Fig. 4b). Root development was also thermoresponsive in the *prr7* mutant, but this response was weaker than the response seen in WT or the *elf3* mutant (Fig. 4a and b). As with the *elf3* mutant, root development in the *prr7* mutant also did not show a differing response between the constant 28°C or

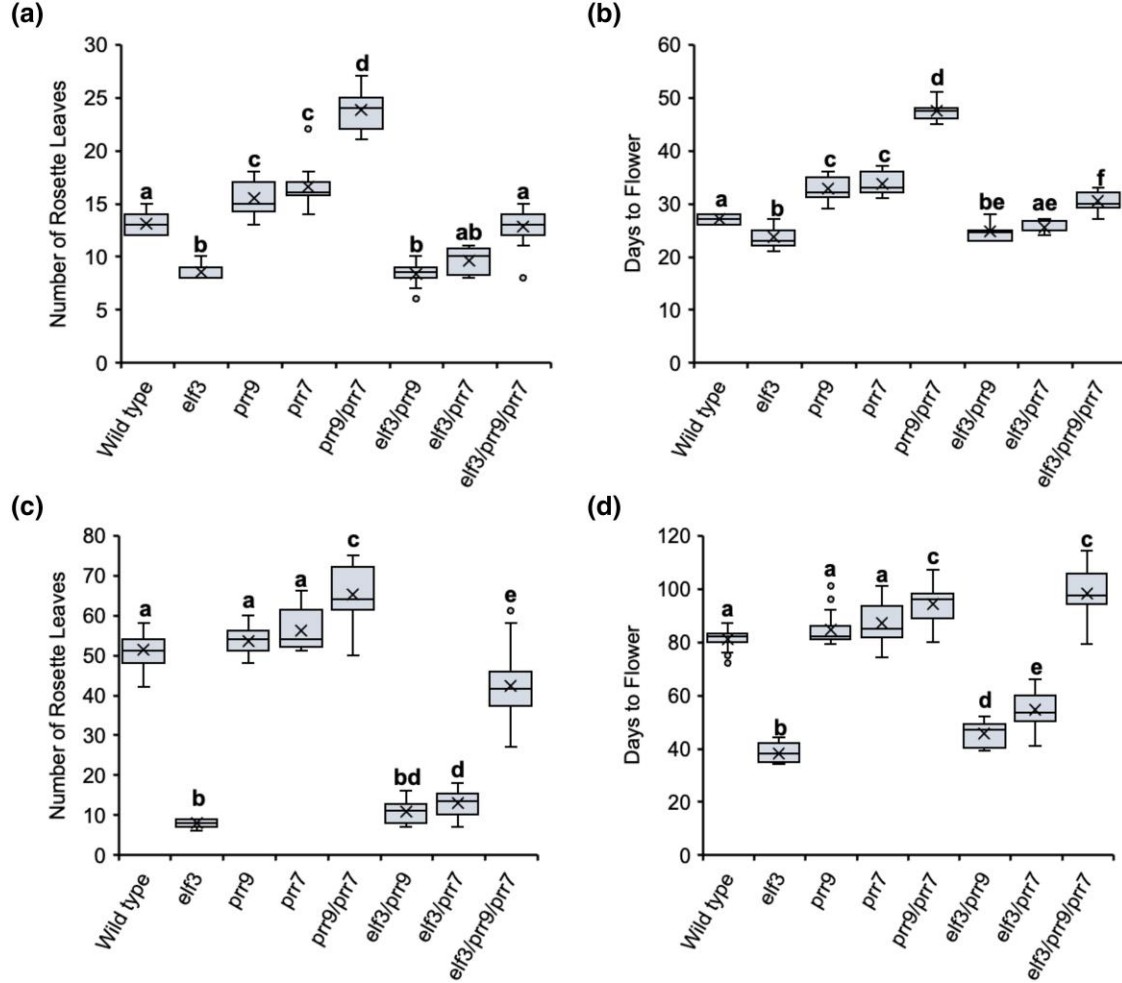

**Fig. 5.** Mutations in *prr9/prr7* delay the *elf3* flowering phenotype under LD and SD photoperiods. Flowering time in the WT (Col-0), *elf3*, *prr*, and *elf3/prr* mutants under LDs a, b) or SDs c, d). Flowering was measured under each photoperiod by the number of rosette leaves a, c) or the days taken to produce a bolt that was ~1 cm above the rosette b, d). For both photoperiods, plants were grown under a constant temperature of 22°C. Experiments were repeated twice, with a minimum of 8 plants analyzed for each experimental repeat. Letters signify a significant difference of *P* < 0.05 as determined by ANOVA with a Tukey HSD post hoc test.

warm night only (Fig. 4b). In the *prr9* mutant, the temperature responsiveness of root growth was largely lost, with only a minimal, nonsignificant, response observed under both warm conditions (Fig. 4a and b). Notably, thermoresponsiveness was restored in the *prr9/prr7* double mutant, suggesting a complex regulatory pathway underpinning the *prr9* and *prr7* phenotype. Root growth in the *elf3/prr* double and triple mutants was also thermoresponsive and showed a similar response to WT and the *elf3* single mutant (Fig. 4a and b). However, as with the *elf3* mutant, there was no longer a difference in response between warm nights and constitutive exposure to 28°C (Fig. 4b). Altogether, our results suggest a complex relationship between ELF3 and PRR9/7 in controlling root development in response to warm temperatures.

## *Elf3* photoperiod insensitivity is rescued by the *prr9/7* mutation

ELF3 and PRRs are both critical regulators of the photoperiodic flowering time pathway in *Arabidopsis* and other plant species (Osnato *et al.* 2022). In *Arabidopsis*, mutations in *elf3* and *prr9/7* lead to opposite effects on flowering time; *elf3* mutants are early flowering and photoperiod insensitive (Zagotta *et al.* 1996), while the *prr9/7* retains photoperiod sensitivity but flowers late under LD and SD (Nakamichi *et al.* 2007). This differing response between

the *elf3* and *prr9/7* mutant led us to explore potential epistatic interactions in flowering time under LD and SD.

Under LD, the *elf3* mutant flowered slightly earlier than WT plants, while the *prr9* and *prr7* single mutants had a moderate late-flowering phenotype (Fig. 5a and b). This late-flowering phenotype was enhanced in the *prr9/prr7* double mutant, which flowered much later than respective single mutant (Fig. 5a and b). Introducing the *prr9* mutation into the *elf3* background resulted in a similar response as to the *elf3* single mutant. For the *elf3/prr7* mutant, flowering time trended toward a WT response for both measurements of flowering (Fig. 5a and b). However, there was no significant difference between *elf3* and *elf3/prr7* in the number of leaves produced. In the *elf3/prr9/prr7* background, flowering was delayed relative to WT as measured by days to flower but not leaf count (Fig. 5a and b). Together, these results indicate that PRR9/7 and ELF3 likely regulate flowering time under LD through partially overlapping pathways, with PRR9/7 functioning downstream of ELF3.

Under SD, *elf3* mutants flowered extremely early (Fig. 5c and d). Neither the *prr9* nor the *prr7* mutant had a flowering-time phenotype under SD, with both mutants flowering at the same time as WT. Flowering was delayed in the *prr9/prr7* double mutant, suggesting genetic redundancy between PRR9/7 in regulating

flowering under SD photoperiods (Fig. 5c and d). Neither the *prr9* nor *prr7* mutation alone was able to fully restore the *elf3* phenotype to a WT response under SD photoperiods, with both *elf3/prr9* and *elf3/prr7* mutants still flowering earlier than WT plants for both days to flower and the number of leaves (Fig. 5c and d). The phenotype for *elf3/prr9/prr7* mutant was complex, with a disconnect between leaf number and days to flower (Fig. 5c and d). As measured by days to flower, the *elf3/prr9/prr7* triple mutant resembled the *prr9/prr7* double mutant. However, when a number of leaves were measured, the *elf3/prr9/prr7* had fewer leaves than WT. Therefore, this analysis suggests a wider metabolic or plastochron phenotype in the *elf3/prr9/prr7* background.

## Discussion

PRR9 and PRR7 are direct targets of ELF3 through the activity of the EC (Herrero *et al.* 2012; Tong *et al.* 2020), and as a result, the expression of *PRR9* and *PRR7* is constitutively upregulated in the *elf3* mutant background (Kolmos *et al.* 2011). However, the consequence of this misexpression has so far not been tested. Here, we utilized mathematical modeling to simulate the effect of the *prr9* and *prr7* mutations on the circadian phenotype of the *elf3* mutant. Our modeling approach was designed to seek qualitative insights on the presence or absence of rhythmicity in the *elf3/prr9/prr7* mutant background. Therefore, we chose DC2016 model to build our models' structures as DC2016 model was designed to characterize the plant clock at a qualitative level. This is in line with similar approaches that have been previously published. This is in line with previous work using the DC2016 model (De Caluwé *et al.* 2017) and other work on minimal models of the plant circadian clock (Avello *et al.* 2021).

The outputs of the modified DC2016 models used here (Fig. 1a) indicated a possible role for *PRR9* and *PRR7* in contributing to *elf3* arrhythmicity (Fig. 1b and c). However, follow-up experiments analyzing the circadian phenotypes of the different *elf3/prr* mutants through gene expression analysis (Fig. 2) or luminescence reporter (Supplementary Fig. 4) revealed that all combinations of the *elf3/prr9/prr7* mutants were arrhythmic. It is unclear why the mathematical models and our in planta data produced different outputs. PRR9 and PRR7 are transcriptional repressors that are part of a gene family that includes *TOC1*, *PRR3*, and *PRR5*. So far, the EC has been demonstrated to directly repress the expression of *TOC1*, as well as *PRR9* and *PRR7* (Herrero *et al.* 2012; Lee *et al.* 2019). Whether the EC represses *PRR3* or *PRR5* expression remains to be investigated. It is possible that the misexpression of additional *PRRs* in the *elf3* background also contributes to the circadian arrhythmia phenotype and further *prr* mutations are necessary alongside the *prr9/prr7* mutations.

Alternatively, limitations within the implemented models may be responsible for the discrepancies we have observed. To understand this possibility, we performed additional modeling with the previously described U2019.3 model (Urquiza-García and Millar 2021) as this model differs in network structure from the compact DC model that we initially used here (Fig. 1; Supplementary Figs. 1 and 2). The U2019.3 model does not incorporate the direct repressive effect of CCA1 and LHY on *PRR9* and *PRR7* (Adams *et al.* 2015) and the negative autoregulation effect of TOC1 included in the latest mathematical models of the plant circadian clock (Huang *et al.* 2012). It also has additional components and a larger number of parameters than the DC2016 model. Simulating the *elf3/prr9/prr7* mutation in the U2019.3 model revealed that the *elf3/prr9/prr7* triple mutant was arrhythmic for both the CCA1/LHY and TOC1 output (Supplementary Fig. 3), supporting our experimental

results (Fig. 3; Supplementary Fig. 4). This paradoxical result highlights the importance of the combined work of modeling and experiments. In future work, it will be necessary to focus further modeling efforts on understanding these discrepancies between models.

Alongside investigating the effect of the *prr9/prr7* mutation on the *elf3* circadian phenotype, we also characterized the *elf3/prr* developmental phenotypes. We firstly investigated hypocotyl development, where we found that the *elf3/prr* combinations closely resembled the *elf3* phenotype and there was no enhancement of the *elf3* phenotype (Fig. 3). This suggests that PRR9/7 function upstream of ELF3 in the same genetic pathway to control hypocotyl development. Circadian regulation of hypocotyl development primarily occurs through the control of PIF activity. ELF3 regulates the expression of *PIF4*, *PIF5*, and *PIF7* through the EC, while independently also controlling PIF4 transcriptional activity (Nusinow *et al.* 2011; Nieto *et al.* 2015; Jiang *et al.* 2019). Separately, PRR9 and PRR7 regulate *PIF4* expression, while also directly inhibiting PIF4 functional activity (Liu *et al.* 2016; Martín *et al.* 2018). Thus, it was unsurprising that there is no further enhancement of the *elf3* phenotype in the *elf3/prr9/prr7* background as *PIF4* expression and activity are already enhanced in the *elf3* background.

We also observed that the *elf3* and *prr* single mutants were equally compromised in root development (Fig. 4). This short-root phenotype was not enhanced in the *prr9/prr7* or *elf3/prr* double mutants, with all lines displaying similar phenotypic defects as the respective single mutants. However, the *elf3/prr9/prr7* triple mutant had a shorter root than the respective single mutants, indicating that ELF3 and PRR9/7 independently control root development (Fig. 4). Supporting a role for these genes in controlling development, *ELF3*, *PRR9*, and *PRR7* are all expressed in the seedling root, albeit less than in the hypocotyl or mature leaf (Supplementary Fig. 6; Klepikova *et al.* 2016). So far, few studies have investigated how the circadian clock controls root development. PRR9/7 may regulate root development via controlling TOR signaling (Li *et al.* 2019), but further studies are needed to understand how the circadian clock controls root development.

Exposure to constitutively elevated temperature promotes hypocotyl and root development (Quint *et al.* 2016; Lee *et al.* 2021). Here, we observed that elevated temperatures during the night were also sufficient in promoting hypocotyl and root development in WT plants. However, hypocotyl and root development more strongly responded to constant exposure to warm temperatures than nighttime only exposure to warm temperature (Figs. 3 and 4). ELF3 was necessary for elongation of the hypocotyl in response to warm temperatures regardless of the duration of temperature exposure (Fig. 3b), supporting previous work (Box *et al.* 2015; Raschke *et al.* 2015; Zhang *et al.* 2021). In contrast, root development remained sensitive to temperature in the *elf3* mutant background, but there was no difference between different durations of warmth exposure (Fig. 4b). Together, our results support recent work that highlighted shoot thermosensors may not function as root thermosensors, and thermomorphogenesis in shoots and roots uses different genetic pathways (Lee *et al.* 2021; Borniego *et al.* 2022; Ai *et al.* 2023).

Hypocotyl elongation in *prr9* and *prr7* remained temperature responsive under both constant conditions and warm nights only, although there was a smaller magnitude difference between the 2 different conditions in the *prr7* mutant (Fig. 3b). The temperature responsiveness of the *prr9/prr7* double mutant was further impaired, and there was no magnitude difference between the different warm temperature regimes, indicating a redundant role for PRR9/PRR7 in mediating daytime hypocotyl thermomorphogenesis (Fig. 3b). TOC1 regulates the time-of-day response to

warm temperature by directly binding to and subsequently inhibiting the activity of PIF4 (Zhu *et al.* 2016). This mechanism ensures thermomorphogenesis is phased to occur only in the late evening and early morning. Under ambient temperatures, PRR9 and PRR7 directly interact with PIF4 to control the timing of *CYCLING DOF FACTOR6* (CDF6) expression, a positive regulator of hypocotyl elongation (Martín *et al.* 2018). Whether similar mechanisms control hypocotyl thermomorphogenesis remains to be investigated.

In the roots, our results suggest a complex genetic pathway underpins the response to warm temperature response. Root development in the *prr9* mutant was insensitive to warm temperature regardless of the length of warmth exposure (Fig. 4b). This response was then fully rescued in the *prr9/prr7*, *elf3/prr9*, and *elf3/prr9/prr7* mutant combinations. So far, no studies have highlighted a role for PRR9 in controlling root thermomorphogenesis and the pathway(s) regulating root thermomorphogenesis remain unclear. A recent study has revealed that root thermomorphogenesis is mediated by an unidentified regulator that controls auxin-dependent progression of the cell cycle (Ai *et al.* 2023). This signaling pathway occurs independently of the well-described PIF thermal integration pathway that regulates thermomorphogenesis in aerial tissue (Delker *et al.* 2022; Ai *et al.* 2023). Thus, it is unclear how the circadian clock regulates root thermomorphogenesis. TOC1, another PRR within the circadian clock, has been described to regulate cell cycle progression in leaf tissue (Fung-Uceda *et al.* 2018). Alternatively, all stages of auxin homeostasis and signaling are under circadian control (Covington and Harmer 2007). Further studies are needed to investigate the integration point for the circadian clock in controlling root thermomorphogenesis.

The regulatory connection between ELF3 and PRRs has emerged as a critical node in the photoperiodic control of flowering time in agronomically important crops (Faure *et al.* 2012; Zhao *et al.* 2012; Alvarez *et al.* 2023; Woods *et al.* 2023). In *Arabidopsis*, the importance of the *PRRs* in contributing to the photoperiod-insensitive early flowering of the *elf3* mutant has remained untested. Under LD, introducing the *prr9/prr7* mutation into the *elf3* background rescued the early flowering *elf3* phenotype and resulted in a mild late-flowering phenotype when measured by days to flower (Fig. 5b) but only a WT response when measured by leaf count (Fig. 5a). These discrepancies were further enhanced under SD. When measured by days to flower, the *elf3/prr9/prr7* triple mutant resembled the late flowering *prr9/prr7* double mutant (Fig. 5d). However, by leaf count, the *elf3/prr9/prr7* triple mutant had fewer leaves than WT (Fig. 5c), thus technically an early flowering plant. Together, these results may highlight a plastochron phenotype in the *elf3/prr9/prr7* mutant. Plastochron regulation is a complex trait, with multiple signaling pathways contributing to the timing of leaf emergence (Werner *et al.* 2001; Wang *et al.* 2008; Lohmann *et al.* 2010; Mimura *et al.* 2012; Cornet *et al.* 2021). One regulator of the plastochron is gibberellin (GA) signaling (Mimura *et al.* 2012; Mimura and Itoh 2014). In barley, *elf3* mutants overaccumulate GA and this misregulation contributes to the *elf3* early flowering under noninductive SD photoperiods (Boden *et al.* 2014). It will be of interest to investigate whether GA is a contributory factor to the complex flowering time phenotypes observed in the *elf3/prr9/prr7* mutant background.

## Conclusion

The *elf3* circadian arrhythmicity was first described nearly 30 years ago (Hicks *et al.* 1996), but we still do not understand the causative factors. In this study, we hypothesized that misregulation of *PRR9* and *PRR7* may explain the *elf3* arrhythmicity.

Although our mathematical modeling supports such a hypothesis (Fig. 1), we could not replicate such results in vivo (Fig. 2; Supplementary Fig. 3). Further work is needed to untangle whether further genetic redundancy among the *PRRs* is responsible for the discrepancy between the simulations and in vivo results or if the differing results are caused by limitations within the model structure used.

## Data availability

All data associated with this manuscript, including the modeling files, are available in the Zenodo repository: https://doi.org/10.5281/zenodo.10115364.

Supplemental material available at GENETICS online.

## Funding

This work was supported by funding from the Biotechnology and Biological Sciences Research Council (BBSRC): SJD (BB/N018540/1), DE (BB/V006665/1), and DE (BB/S506795/1). We also acknowledge BBSRC Whiterose DTP studentship: JR (BB/M011151/1, ref 1792522) and EJR (BB/T007222/1, ref 2444228). DE was also supported by the Royal Society (RGS\R2\212345). This work was also supported by the National Natural Science Foundation of China to QX (32170259) and XX (U1904202 and 32170275) and the National Key Research and Development Program of China to XX (2021YFA1300402). Deutsche Forschungsgemeinschaft (Qu 141/12-1) and the European Social Fund and the Federal State of Saxony-Anhalt (International Graduate School AGRIPOLY—Determinants of Plant Performance, grant no. ZS/2016/08/80644) supported MQ. PF was supported by the CONICYT PFCHA/DOCTORADO BECAS CHILE award (2013–72140562).

## Conflicts of interest

The author(s) declare no conflicts of interest.

## Author contributions

Experimental design: PA, AMD, DE, JWP, MQ, QX, XX, SJD, and JR. Data generation: LY, PA, ZZ, SCLL, KM, EJR, AMD, YS, and JR. Data analysis: all authors. Manuscript writing: SJD and JR. Manuscript edits: all authors.

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

*Editor: T. Juenger*