## [Peer Review File · Genetics]

Complex epistatic interactions between ELF3, PRR9, and PRR7 regulates the circadian clock and plant physiology

Li Yuan, Paula Avello, Zihao Zhu, Sarah Lock, Kayla McCarthy, Ethan Redmond, Amanda Davis, Yang Song, Daphne Ezer, Jonathan Pitchford, Marcel Quint, Qiguang Xie, Xiaodong Xu, Seth Davis, and James Ronald

NOTE: The reviews and decision letters are unedited and appear as submitted by the reviewers.

In extremely rare instances and as determined by a Senior Editor or the EIC, portions of a review may be redacted. If a review is signed, the reviewer has agreed to no longer remain anonymous.

The review history appears in chronological order.

Review Timeline:

Submission Date:	2023-08-07
Editorial Decision:	2023-09-08
Resubmission Received:	2023-11-14
Accepted:	2023-12-05

September 8, 2023

GENETICS-2023-306407

Complex epistatic interactions between ELF3, PRR9, and PRR7 regulates the circadian clock and plant physiology

Dear Dr. Davis:

Two experts in the field have reviewed your manuscript, and I have read it as well. The editor and reviewers were enthusiastic about your approach linking mathematical models of circadian pathways with molecular genetic tests of hypotheses. While your manuscript is not currently acceptable for publication in GENETICS, we would welcome a substantially revised manuscript. Both reviewers have comments and concerns to be addressed in a revised manuscript. You can read their reviews at the end of this email.

The motivation for the current study stems from model predictions on clock regulatory dynamics and plant growth and development. The literature in the field is quite advanced, with a number of existing models and model predictions. Moreover, there are many empirical studies of mutants including studies of epistatic interactions among key elements. This study focuses specifically on ELF3 regulation of PRR genes. One limitation of the study is that it lacks clear motivation or justification for use of the C2016 model (amongst other possibilities), there is little discussion of its limitations, and some relevant citations are missing. Reviewer 2 argues that a potentially more realistic model (U2019.3 or U2020.3) would be more appropriate for generating hypotheses and for comparison to the empirical work presented. The reviewer argues that these models would be more powerful compared to the predictions of the "compact model". The manuscript would be strengthened if these issues can be addressed. Both reviewers had questions about the statistical results and their interpretation. It would be helpful to include more detail about the statistical tests performed, including adding p-values in text and figures when appropriate. Be careful that the interpretation and conclusions are consistent with the evidence. Finally, the reviewers suggest a number of additional experiments that could improve the experiment, including new gene expression studies or manipulation of TOC1. While the editor agrees that these experiments would strengthen the paper, adding new empirical work to the existing study is not a requirement for consideration of a resubmission.

Upon resubmission, please include:

1. A clean version of your manuscript;
2. A marked version of your manuscript in which you highlight significant revisions carried out in response to the major points raised by the editor/reviewers (track changes is acceptable if preferred);
3. A detailed response to the editor's/reviewers' feedback and to the concerns listed above. Please reference line numbers in this response to aid the editor and reviewers.

Your paper will likely be sent back out for review.

Additionally, please ensure that your resubmission is formatted for GENETICS

<https://academic.oup.com/genetics/pages/general-instructions>

Follow this link to submit the revised manuscript: Link Not Available

Sincerely,
Tom Juenger
Series Editor, Plant Genetics & Genomics

Approved by:
David Greenstein
Senior Editor
GENETICS

Reviewer #1 (Comments for the Authors (Required)):

In their manuscript entitled "Complex Epistatic Interactions between ELF3, PRR9, and PRR7 Regulate the Circadian Clock and Plant Physiology", Yuan et al. investigate the modeling and genetic interactions among three clock genes, ELF3, PRR9, and PRR7, in Arabidopsis to test the regulatory dynamics in the circadian clock. The authors simulated the impact of the prr9/prr7 mutation on the elf3 circadian phenotype. They also investigated epistatic interactions among three clock genes using single, double, and triple mutants across different growth and development processes.

This is a well-designed manuscript with a clear hypothesis. And the entire manuscript is globally easy to follow. The results of the study are not entirely unique. The mathematical model for these core circadian clock genes has previously been explored, as referenced by the authors (De Caluwé et al., 2016). Furthermore, the genetic interactions among these three genes in similar phenotypes have also been reported (though not included in the reference; please refer to the below sources). However, the current study aims to separate the roles of the PRR9 and PRR7 genes within the model and elucidate the complex epistatic interactions involving all three genes across diverse phenotypic outcomes. My comments on the manuscript are listed below. It is a bit challenging to review this manuscript since there are no line numbers in the merged file, but I will do my best to make it clear.

1) The ELF3, PRR9, and PRR7 are well-studied genes. The manuscript should include a comparative analysis, discussion, and appropriate referencing from prior literature. A few examples that are closely related to this manuscript, but are not found in the reference, are provided below:

PSEUDO-RESPONSE REGULATORS, PRR9, PRR7, and PRR5, Together Play Essential Roles Close to the Circadian Clock of *Arabidopsis thaliana*. *Plant and Cell Physiology*, 2005 (<https://doi.org/10.1093/pcp/pci086>)

In this paper, the authors used single, double, and triple PRR mutants to investigate circadian-associated phenotypes, including photoperiodic flowering time and the length of hypocotyls.

Ambient temperature response establishes ELF3 as a required component of the core *Arabidopsis* circadian clock. *PNAS*, 2010 (<https://doi.org/10.1073/pnas.0911006107>)

In this paper, the authors investigated the expression change of PRR9/7 in the *elf3* mutant background in response to ambient temperature.

2) There are a lot of pairwise comparisons of a series of phenotypes between mutants and the WT in the results (Figures 3, 4, and 5). For example, on Page 8, the author mentioned that "the *elf3* mutant had a long hypocotyl phenotype under these conditions compared to WT Col-0 (Figure 3A)"; on Page 9, the author mentioned that "Root development was strongly impaired in the *elf3*, *prr9*, and *prr7* single mutants at 20°C compared to WT, with each respective single mutant having a similar response to each other (Figure 4A). There was no change in the primary root length of *prr9/prr7* double and *elf3/prr* double combinations compared to the respective single mutants. However, root development was further impaired in the *elf3/prr9/prr7* triple mutant compared to the single and double mutants (Figure 4A)."... However, the author did not provide the statistical results for those comparisons. Please include p-values throughout Figures 3, 4, and 5 for those comparisons. Additionally, please clarify the statistical methods used in those comparisons in the Methods section (Page 17). RStudio and/or R are not the methods for the statistical analysis.

3) As the authors mentioned on page 9, those circadian genes, such as ELF3 and PRR9/7, may regulate root development at ambient temperatures. I am curious about the expression levels of ELF3, PRR9, and PRR7 genes across different tissues. Incorporating supplementary data of tissue-specific expression profiles from online databases like the Bio Analytic Resource/eFP could provide valuable information in this regard.

4) On page 10, the authors examined the number of rosette leaves and days to flower in WT plants and mutants. I recommend that the authors examine the expression levels of florigen and downstream flowering time genes, including FT, TSF, and SOC1, in those lines. This may help readers in comprehending the link between upstream clock genes and reproductive phenotypes.

5) I'm not an expert in modeling, yet I have an open question to discuss. As the authors mentioned the phenotype of root development in those mutants, did they include tissue as a factor in their modeling? Or, in other words, do the authors think the pattern of circadian rhythm in above-ground (seedlings or leaves) is the same as that in below-ground (roots)?

6) Page 9, "PRR9, but not ELF3, regulates root development under warm temperatures." Typing issue?

Reviewer #2 (Comments for the Authors (Required)):

Comments to the authors:

In this manuscript by Yuan L et al., the authors use a compact mathematical model of the clock (De Caluwe et al., 2016) to shed light on the functional significance of the ELF3 regulation of PRR9 and PRR7. They particularly focused on how this regulation affects circadian clock dynamics and plant development. To tackle this question they first model the effects of a *prr9/prr7* mutation on an *elf3* mutant background. The model they employed suggested that this would rescue the circadian arrhythmicity of *elf3*. They then generated an *Arabidopsis* triple mutant to validate the findings of the model and further tested their circadian and development phenotypes. Their data shows that *elf3* is needed for hypocotyl growth under both, ambient and warm temperatures, while *prr9* is critical for root thermomorphogenesis. Furthermore, the authors conclude that the *prr9* or *prr7* mutations could rescue the photoperiod insensitive early flowering of *elf3*.

Major concern:

The authors used the C2016 compact model of the *Arabidopsis* clock and then split the variables of *prr7* and *prr9*. They performed simulations to model the effects of a *prr9/prr7* mutation on an *elf3* mutant background and found that this would rescue the circadian arrhythmicity of *elf3*. However, they do not observe this experimentally.

The concern here is the model of choice. For example, running a simulation with either one of the two most realistic models that work with absolute units: U2019.3 and U2020.3 (Urquiza and Millar 2020) results in arrhythmicity (see attached images). In particular, in 2019.3 which has a better network architecture than U2020.3, the levels of TOC1 mRNA are around 2x higher than WT. Moreover, the predicted levels of CCA1/LHY are lower. This could result in two effects: 1) on one hand reduction of PIF

activity mediated by TOC1, as it has been shown that TOC1 can suppress PIF3; 2) the reduction in the levels of CCA1/LHY can result in acceleration of flowering. Also, TOC1 has been implicated in root development (<https://doi.org/10.1038/ncomms8641>) Using a mathematical model in absolute units would provide more powerful predictive power/insights than the compact model of C2016. C2016 lacks this capacity as the scale is compressed to the interval [0,1] and therefore lacks the ability to propose the effect of changing RNA levels.

The U2019.3 model suggests that the effect on hypocotyl length reduction in the triple mutant could potentially be attributed to high levels of TOC1 in combination with low levels of CCA1/LHY. This hypothesis would then need to be further verified by the authors. I would suggest abandoning the C2016 model and switching to the U2019.3. Then, verifying the predictions of high levels of TOC1, and introducing a TOC1 mutation (by CRISPR, for example) into the triple *prp9/prp7/elf3* mutant to remove the suppression of hypocotyl and observe a delay in flowering.

Other concerns:

Results section:

In the results section you make the following statement:

"Both the *prp9* and *prp7* mutations were able to partially rescue the *elf3* early flowering phenotype under SD and, as with LD, the *prp7* mutation had a stronger effect on rescuing the *elf3* phenotype than *prp9* (Figure 5)."

However,

Fig. 5A (LD - rosette leaves): the double *elf3/prp7* and *elf3/prp9* mutants are statistically the same as the single *elf3* mutant.

Fig. 5B (LD - days to flowering): the double *elf3/prp9* mutants are statistically the same as the single *elf3* mutant. Here, only the double *elf3/prp7* is similar to wild type.

Fig. 5C (SD - rosette leaves): the double *elf3/prp9* mutants are statistically the same as the single *elf3* mutant. The double *elf3/prp7* is different to both *elf3* and wild type. The triple mutant does not fully rescue the phenotype.

Fig. 5D (SD - days to flowering): the double *elf3/prp7* and *elf3/prp9* mutants partially rescue the flowering phenotype of the *elf3* mutant.

Please revise your statement accordingly to reflect the data more accurately. Or let the readers know that you specifically refer to the 'days-to-flowering' data.

Methods section:

What is the light intensity of constant red and blue light you used for the TopCount experiments? Please add this to the 'Luciferase circadian experiments' subsection.

Data Availability Statement:

The Data Availability Statement does not indicate where the readers can access the data to support the findings. It is not enough nowadays to simply state that 'data are available upon request'. Please try to ensure that when you have the final version of the manuscript, you have deposited the data in a public repository with a persistent identifier.

Associate Editor Comments:

We thank both reviewers for their positive comments on our manuscript and we thank the editor for providing us the opportunity to respond to their comments below. The line numbers below refer to the clean manuscript file.

Reviewer 1

The results of the study are not entirely unique. The mathematical model for these core circadian clock genes has previously been explored, as referenced by the authors (De Caluwé et al., 2016). Furthermore, the genetic interactions among these three genes in similar phenotypes have also been reported (though not included in the reference; please refer to the below sources).

We respectfully disagree with the reviewer. Although the work of De Caluwé *et al.*, (2016) designed and parameterised the compact model that we have used as the basis for our work, they did not carry out simulations of the *elf3/prr* mutant combinations as we have done. Similarly, there has been no research to date on the epistatic relationship of *elf3/prr9/prr7* mutant in regulating circadian rhythms or plant physiology. Our study is the first to generate and characterise these different mutant combinations. Additionally, our work sheds new light on the clock network structure currently hypothesised in mathematical models based on ordinary differential equations. Thus, we believe that this makes our work novel.

1) The ELF3, PRR9, and PRR7 are well-studied genes. The manuscript should include a comparative analysis, discussion, and appropriate referencing from prior literature. A few examples that are closely related to this manuscript, but are not found in the reference, are provided below.

We thank the reviewer for highlighting this. We had previously included a detailed discussion on ELF3 and the evening complex (line 77-93), the overlap and distinguishments between PRR9 and PRR7 in the circadian clock (line 95-97 and line 99-105) and the role of ELF3/EC in repressing *PRR9* and *PRR7* expression (line 105 – 111) with appropriate referencing of the literature. We have included the two references as requested by the reviewer. The first reference is included on line 99, while the second reference is included on line 106. For both new references, we have added additional sentences to provide context (line 97-99 and line 105 – 106, respectively).

2) However, the author did not provide the statistical results for those comparisons. Please include p-values throughout Figures 3, 4, and 5 for those comparisons.

Where appropriate, we had indicated significance difference on all graphs (Figure 3-5) as letters, with different letters indicating significant difference between the genotypes and growth conditions. This is standard practice for datasets where comparisons are being made between multiple genotypes and/or growth conditions. We also included in the figure legend of each figure the statistical test and the alpha value ($p < 0.05$). We do not think it is feasible

to provide the p-values in a readable format because of the large number of p-values associated with each graph. For example, figure 3 alone has over 250 individual p-values due to the large number of genotypes and multiple growth conditions

Additionally, please clarify the statistical methods used in those comparisons in the Methods section (Page 17). RStudio and/or R are not the methods for the statistical analysis.

We have expanded the methods to include a description of the statistical test used for each experiment. This is included at the end of the manuscript (Line 545-549).

3) As the authors mentioned on page 9, those circadian genes, such as ELF3 and PRR9/7, may regulate root development at ambient temperatures. I am curious about the expression levels of ELF3, PRR9, and PRR7 genes across different tissues. Incorporating supplementary data of tissue-specific expression profiles from online databases like the Bio Analytic Resource/eFP could provide valuable information in this regard.

We thank the reviewer for this suggestion. We have included expression profiles for *ELF3*, *PRR9* and *PRR7* in hypocotyl, root and mature leaf tissue as Supplementary Figure 6. This data was taken from the Klepikova RNAseq expression atlas. We have discussed this data in the discussion and included appropriate referencing (line 372-374).

4) On page 10, the authors examined the number of rosette leaves and days to flower in WT plants and mutants. I recommend that the authors examine the expression levels of florigen and downstream flowering time genes, including FT, TSF, and SOC1, in those lines. This may help readers in comprehending the link between upstream clock genes and reproductive phenotypes.

We thank the reviewer for this suggestion, and we agree that it would be interesting to understand the gene expression dynamics of floral regulators during flowering in these different mutant backgrounds. However, for WT and some of the mutant lines, flowering can take over three months under a short-day photoperiod. This also does not factor in the time needed to carry out the subsequent experimental work. Thus, we believe that this beyond the scope of this work.

5) I'm not an expert in modelling, yet I have an open question to discuss. As the authors mentioned the phenotype of root development in those mutants, did they include tissue as a factor in their modelling? Or, in other words, do the authors think the pattern of circadian rhythm in above-ground (seedlings or leaves) is the same as that in below-ground (roots)?

We thank the reviewer for this discussion point. The model we employed does not distinguish tissue type. As we have highlighted in the introduction (line 86-87) and in the discussion (line 374-377), the structure of the plant circadian clock in root tissues is still not established. With

the sparsity of studies on the root circadian clock (and the conflicting results between the few studies that have been complete (For example: James *et al.*, 2008, Bordage *et al.*, 2016, Chen *et al.*, 2020, Nimmo *et al.*, 2020) it is difficult to discuss whether the shoot and root clocks have different genetic structures without a better understanding of the root clock.

6) Page 9, "PRR9, but not ELF3, regulates regulate root development under warm temperatures." Typing issue?

We thank the reviewer for highlighting this. We were trying to highlight that root development in the *prp9* mutant background is completely insensitive to warm temperatures. We have amended this sub-title (line 241) to better reflect this.

Reviewer #2 (Comments for the Authors (Required)):

1) The authors used the C2016 compact model of the Arabidopsis clock and then split the variables of *prp7* and *prp9*. They performed simulations to model the effects of a *prp9/prp7* mutation on an *elf3* mutant background and found that this would rescue the circadian arrhythmicity of *elf3*. However, they do not observe this experimentally.

The concern here is the model of choice. For example, running a simulation with either one of the two most realistic models that work with absolute units: U2019.3 and U2020.3 (Urquiza and Millar 2020) results in arrhythmicity (see attached images). In particular, in 2019.3 which has a better network architecture than U2020.3, the levels of TOC1 mRNA are around 2x higher than WT. Moreover, the predicted levels of CCA1/LHY are lower. This could result in two effects: 1) on one hand reduction of PIF activity mediated by TOC1, as it has been shown that TOC1 can suppress PIF3; 2) the reduction in the levels of CCA1/LHY can result in acceleration of flowering. Also, TOC1 has been implicated in root development (<https://doi.org/10.1038/ncomms8641>).

We thank the reviewer for suggesting the use of the U2019.3 model. We have implemented this model and have added the results in Supplementary material (Figure 3) along with text in Discussion (line 338-351), and Methods sections (lines 488-491). Although we believe that Urquiza and Millar 2020 have introduced an interesting approach to mathematical modelling, we respectfully disagree with the reviewer that U2019.3 is a more realistic models of the plant circadian clock. Below, we discuss our reasoning:

- **The network architecture.** U2019.3 model does not include the latest empirical discoveries for the plant clock (ODEs in Supplementary information, Urquiza and Millar 2020), namely, the direct repressive effect of CCA1 and LHY on *PRR9* and *PRR7* (Adams *et al.* 2015), and the autoregulation effect of TOC1 (Huang *et al.* 2012). To support this point, we have now added the following reference to the main text:

Huang, W., Pérez-García, P., Pokhilko, A., Millar, A., Antoshechkin, I., et al. (2012) Mapping the core of the Arabidopsis circadian clock defines the network structure of the oscillator. *Science*, 336: 75–79.

- **The data are log-transformed for the optimization process.** Parameter estimation for U2019.3 model was obtained from log-transformed data, and a comparison between model's outputs and experimental data is shown in log base 10 scale, which makes it challenging to fairly assess the fit of the model to the actual data.
- **Constraints on RNA levels.** Parameter estimation for U2019.3 model was obtained by constraints on experimental data, period, and amplitude. The latter was necessary to permit sustained oscillations in constant light (Urquiza and Millar 2021). Note that the scaling factors have a value of 1; the multiplicative neutral (table 2, Urquiza and Millar 2022, Supplementary information). It is not clear whether the constrains on amplitude could explain the results observed for *elf3* in this model, and further numerical investigation would be needed, which we believe is beyond the scope of our investigation. The following references were added to support this:

Urquiza-García, U. & Millar, A.J. (2021) Testing the inferred transcription rates of a dynamic, gene network model in absolute units. *In Silico Plants*, 3, 1–18.

Pokhilko, A. et al. (2012) The clock gene circuit in Arabidopsis includes a repressilator with additional feedback loops. *Mol. Syst. Biol.* 8, 574.

2.1) Using a mathematical model in absolute units would provide more powerful predictive power/insights than the compact model of C2016.

We thank the reviewer for helping us clarify this point. We agree with reviewer 2 that using a mathematical model in absolute units would be the best option for contrasting simulated data against experimental data that is also in absolute units. This takes relevance in situations when quantitative insights on mRNA point predictions are sought. However, our modelling approach here is different, and we have added text to clarify this (line 319-323).

2.2) C2016 lacks this capacity as the scale is compressed to the interval [0,1] and therefore lacks the ability to propose the effect of changing RNA levels.

We respectfully disagree with the reviewer. We acknowledge that in De Caluwé *et al.*, 2016, the model's outputs and the experimental data were transformed to lay in the range of 0 and 1 to allow a comparison between their mRNA waveforms. This was obtained via normalisation by dividing each point of the data by their respective maximum. This forced the model's variables to have a maximum value of 1 for the results displayed in De Caluwé *et al.*, 2016; however, the model is not restricted to the interval [0, 1]. For the optimisation process, the constraints on amplitude in the DC2016 model required all variables of the model to have a

minimum value of 0.1, and that the difference between their minimum and maximum values were at least of 10%. The constraints were defined for wild type in continuous light and darkness, and for all mutants in continuous light (De Caluwé *et al.*, 2016). This allows the model to be capable of generating outputs beyond the range [0, 1] (De Caluwé *et al.*, 2017). There was not a constraint limiting variable values between 0 and 1. The other constraints used in the De Caluwé *et al.*, 2016 model were on phase (CCA1/LHY peaking between ZT22 and ZT4), and on period length (e.g., between 24 and 25 hours in constant light for wild type).

Here we followed the approach in De Caluwé *et al.*, 2017 (please see response 2.1), where the authors analysed the dynamics of the model in details without normalising the simulated data nor the experimental data (See figures 2, 3, 5, 8, and 9 in <https://doi.org/10.1016/j.jtbi.2017.03.005>). In De Caluwé *et al.*, 2017, simulations and experiment outputs were compared on an oscillatory behaviour basis relative to the entrainment. We, therefore, did not normalise the simulated data, but we kept the original values of the model's variables. Also, to simulate mutant observations, the authors in De Caluwé *et al.*, 2016 proposed to change the mRNA levels by decreasing relevant rate constants of transcription; we followed the same approach (line 488-491).

3.1) The U2019.3 model suggests that the effect on hypocotyl length reduction in the triple mutant could potentially be attributed to high levels of TOC1 in combination with low levels of CCA1/LHY. This hypothesis would then need to be further verified by the authors. I would suggest abandoning the C2016 model and switching to the U2019.3.

This is an insightful point. We thank the reviewer for the suggestion; however, we have respectfully preferred not to abandon the DC2016 model to replace it with U2019.3 for two main reasons. First, U2019.3 is not aligned with the latest biological findings of the plant clock (please see comment 1 above), and second, we sought qualitative insights, so that we used a model designed for this purpose (please see response to comment 2.1 above). Nonetheless, we have now included results from U2019.3 model (Supplementary Figure 3) and comments in the discussion (line 338-351) as we mentioned in response to question one.

3.2) Then, verifying the predictions of high levels of TOC1, and introducing a TOC1 mutation (by CRISPR, for example) into the triple prr9/prr7/elf3 mutant to remove the suppression of hypocotyl and observe a delay in flowering.

We agree with the reviewer that high levels of *TOC1* could be responsible for some of the complex phenotypes. As we highlight in the discussion it is already known in the literature that the *elf3* mutant would have high levels of *TOC1* as ELF3/EC represses *TOC1* expression (Lee *et al.*, 2019 – described on line 333-334). However, we believe that verifying this experimentally is beyond the scope of the current paper. Generating and validating the mutant as requested (either through CRISPR or crossing the *toc1* single mutant into the triple mutant

background) combined with carrying out the proposed experiments would take more than 12 months of work. We believe such work is beyond the scope of this paper.

4) In the results section you make the following statement: “Both the *prp9* and *prp7* mutations were able to partially rescue the *elf3* early flowering phenotype under SD and, as with LD, the *prp7* mutation had a stronger effect on rescuing the *elf3* phenotype than *prp9* (Figure 5).”

We thank the reviewer for highlighting this and accordingly we have amended the statement on line 303 to better reflect the data. During the re-analysis of the datasets, we realised that there was an error for the letters attributed to *elf3/prp9/prp7* on Figure 5C (it should have been E, not D as originally described). We apologise for this error, and we have corrected them in the latest version of the figures. These changes do not affect the conclusion of the results or discussion. We have also checked all other significance values, with no other errors found.

Methods:

5) What is the light intensity of constant red and blue light you used for the TopCount experiments? Please add this to the 'Luciferase circadian experiments' subsection.

This has been added to **line 508** of the text.

6) The Data Availability Statement does not indicate where the readers can access the data to support the findings. It is not enough nowadays to simply state that 'data are available upon request'. Please try to ensure that when you have the final version of the manuscript, you have deposited the data in a public repository with a persistent identifier.

We thank the reviewer for highlighting this. We have created a folder with Zenodo and we are in the process of uploading the raw data and modelling files to this repository. The data sharing conclusion (**line 551-553**) has been appropriately modified to reflect this.

December 5, 2023

RE: GENETICS-2023-306643

Prof. Seth J. Davis
University of York
Biology
Wentworth Way
York
United Kingdom

Dear Dr. Davis:

Congratulations! We are delighted to inform you that your manuscript entitled "**Complex epistatic interactions between ELF3, PRR9, and PRR7 regulates the circadian clock and plant physiology**" is acceptable for publication in GENETICS. Many thanks for submitting your research to the journal.

To Proceed to Production:

1. Format your article according to GENETICS style, as discussed at <https://academic.oup.com/genetics/pages/general-instructions>, and upload your final files at <https://genetics.msubmit.net>.
2. Your manuscript will be published as-is (unedited-as submitted, reviewed, and accepted) at the GENETICS website as an Advanced Access article and deposited into PubMed shortly after receipt of source files and the completed license to publish. Please notify sourcefiles@thegsajournals.org if you do not wish to publish your article via Advanced Access.
3. We invite you to submit an original color figure related to your paper for consideration as cover art. Please email your submission to the editorial office or upload it with your final files. You can submit a small-sized image for evaluation, and if selected, the final image must be a TIFF file 2513px wide by 3263px high (8.375 by 10.875 inches; resolution of 600ppi). Please avoid graphs and small type.
4. Please double check that the link provided for data release is accurate. The zenodo link provided generated an error and I was unable to access the data repository.

If you have any questions or encounter any problems while uploading your accepted manuscript files, please email the editorial office at sourcefiles@thegsajournals.org.

Sincerely,
Tom Juenger
Series Editor, Plant Genetics & Genomics

Approved by:
David Greenstein
Senior Editor
GENETICS

note: Please add jnls.author.support@oup.com and genetics.oup@kwglobal.com (or the domains @oup.com and @kwglobal.com) to your email program's "safe senders" list. You will be contacted by both at various points during the production process.

Review comments (if applicable):

Reviewer #1 (Comments for the Authors (Required)):

I appreciate your response and the improvements you have made. With no more questions on the manuscript, I am recommending its publication.

Reviewer #2 (Comments for the Authors (Required)):

The authors have addressed all my concerns. I welcome this improved version of the manuscript.